# Therapeutic Drug Monitoring of Subcutaneous Infliximab in Inflammatory Bowel Disease—Understanding Pharmacokinetics and Exposure Response Relationships in a New Era of Subcutaneous Biologics

**DOI:** 10.3390/jcm11206173

**Published:** 2022-10-19

**Authors:** Robert D. Little, Mark G. Ward, Emily Wright, Asha J. Jois, Alex Boussioutas, Georgina L. Hold, Peter R. Gibson, Miles P. Sparrow

**Affiliations:** 1Department of Gastroenterology, Alfred Health and Monash University, Melbourne 3004, Australia; 2Department of Gastroenterology, St. Vincent’s Hospital, Melbourne University, Melbourne 3004, Australia; 3Department of Gastroenterology & Clinical Nutrition, Royal Children’s Hospital, Melbourne 3052, Australia; 4Microbiome Research Centre, St. George Hospital, University of New South Wales, Sydney 2217, Australia

**Keywords:** infliximab, CT-P13, subcutaneous, therapeutic drug monitoring

## Abstract

CT-P13 is the first subcutaneous infliximab molecule approved for the management of inflammatory bowel disease (IBD). Compared to intravenous therapy, SC infliximab offers a range of practical, micro- and macroeconomic advantages. Data from the rheumatological literature suggest that subcutaneous CT-P13 may lead to superior disease outcomes in comparison to intravenous infliximab. Existing studies in IBD have focussed on pharmacokinetic comparisons and are inadequately powered to evaluate efficacy and safety differences between the two modes of administration. However, emerging clinical trial and real-world data support comparable clinical, biochemical, endoscopic and safety outcomes between subcutaneous and intravenous infliximab in both luminal Crohn’s disease and ulcerative colitis. Across the available data, subcutaneous CT-P13 provides relative pharmacokinetic stability and higher trough drug levels when compared to intravenous administration. The clinical impact of this observation on immunogenicity and treatment persistence is yet to be determined. Trough levels between the two methods of administration should not be compared in isolation as any subcutaneous advantage must be considered in the context of comparable total drug exposure and the theoretical disadvantage of lower peak concentrations compared to intravenous therapy. Furthermore, target drug levels for subcutaneous CT-P13 associated with remission are not known. In this review, we present the available literature surrounding the pharmacokinetics of subcutaneous CT-P13 in the context of therapeutic drug monitoring and highlight the potential significance of these observations on the clinical management of patients with IBD.

## 1. Introduction

Inflammatory bowel disease (IBD) comprises a group of chronic, immune-mediated disorders including both ulcerative colitis (UC) and Crohn’s disease (CD) [1,2]. Anti-tumour necrosis factor (anti-TNF) biologics such as infliximab and adalimumab are effective in the induction and maintenance of remission in IBD, with primary response rates of 40–70% [3,4,5,6,7,8]. Infliximab, a chimeric IgG1 monoclonal antibody, has over 25 years of post-marketing efficacy and safety data in IBD [9]. After expiry of the patent for originator infliximab, a number of biosimilars have been developed, allowing market competition and consequent cost savings [10,11]. CT-P13 was the first infliximab biosimilar to be approved for UC and CD in Europe and the USA [12]. Intravenous CT-P13 has demonstrated non-inferior pharmacokinetics, efficacy and safety outcomes in both rheumatological conditions [13,14] and IBD [15,16,17,18,19,20,21]. Adherence to intravenous administration is measurable and may be superior to that of self-administered biologic or oral therapy [22,23,24]. Furthermore, drug storage conditions are likely to be more regulated within hospital pharmacy departments in contrast to the variability and inadequacy reported with home-based biologic storage [22,23,25,26]. However, subcutaneous rather than intravenous biologic therapy is preferable to many patients and clinicians [27,28,29,30]. In comparison to intravenous therapy, de-centralising care provision may allow greater patient flexibility, convenience and a range of individual and macroeconomic healthcare benefits [31,32,33,34,35]. The COVID-19 pandemic highlighted the advantages of transitioning care away from the hospital setting [36]. In this context, a number of guidelines have suggested considering the prioritisation of subcutaneous biologics in IBD management [37,38].

The subcutaneous formulation of CT-P13 attained European Medicines Agency (EMA) approval for IBD in 2020 on the basis of two small randomised controlled trials (RCTs) reporting comparable pharmacokinetics, efficacy and safety of subcutaneous CT-P13 as compared to intravenous CT-P13 in both rheumatoid arthritis and active CD [39,40]. A subsequent meta-analysis comparing subcutaneous CT-P13 with historical intravenous infliximab outcomes in patients with moderate-severe rheumatoid arthritis suggests a potential efficacy benefit with subcutaneous therapy [41]. Similarly, a network meta-regression pooling individual patient data from two RCTs comparing subcutaneous and intravenous CT-P13 in rheumatoid arthritis patients supports subcutaneous CT-P13 providing superior clinical response and improvements in functional disability [42]. The available interventional controlled studies of subcutaneous CT-P13 in IBD to date are inadequately powered to assess efficacy and safety differences between the two methods of administration and are limited to outpatient moderate-severe UC and luminal CD. The pivotal RCT evaluating subcutaneous CT-P13 in IBD was conducted in two parts. Part 1, published in abstract form only, was a phase I, open-label dose-finding RCT in 44 patients with active CD. Following intravenous infliximab induction at week 0 and week 2, patients were randomised to receive either standard maintenance intravenous infliximab 5 mg/kg 8-weekly or 120 mg, 180 mg or 240 mg of subcutaneous CT-P13 2-weekly [40]. Part 2 was an open-label, non-inferiority trial involving 131 anti-TNF naïve patients with active CD or UC across 50 centres. Following intravenous induction, patients were randomised to receive maintenance intravenous infliximab 5 mg/kg 8-weekly or subcutaneous CT-P13 at a dose of 120 mg (if <80 kg) or 240 mg (≥80 kg) 2-weekly [43]. The primary outcomes were pharmacokinetic and will be discussed in detail. Consistent with emerging real-world data, there were comparable clinical, biochemical, endoscopic and safety outcomes between subcutaneous and intravenous infliximab formulations [40,43]. Across all indications, trough drug levels are consistently higher in patients receiving subcutaneous CT-P13 than those treated with intravenous infliximab.

However, comparing drug levels at these time points does not adequately reflect total drug exposure between the two formulations. The significance of this observation on pharmacokinetics, disease activity, immunogenicity and treatment persistence in IBD is yet to be determined and requires further studies incorporating therapeutic drug monitoring (TDM). TDM of intravenous infliximab has been shown to be cost-effective and to improve clinical and objective outcomes in IBD [44,45]. There is a well-established exposure-response relationship for intravenous infliximab, with multiple studies having demonstrated that higher trough drug levels are associated with improved patient outcomes [44,45,46,47]. Evidence supporting TDM of other biologics is less robust, particularly for adalimumab, ustekinumab and vedolizumab [46,48].

There is an unmet need to confirm the value of TDM of subcutaneous infliximab, interpret the significance of elevated trough drug levels and determine concentration thresholds associated with remission. Furthermore, the potential disadvantages of lower peak concentrations with subcutaneous therapy requires evaluation. This review aims to appraise the literature surrounding subcutaneous CT-P13 TDM, highlight the current knowledge gaps, and provide guidance for clinical practice.

## 2. Search Strategy

A literature search was conducted using PubMed Online and the Cochrane Library databases. The search was performed using the following linked search terms: ‘CT-P13 OR infliximab;’ AND ‘subcutaneous;’ AND ‘Crohn’s disease (CD) OR ulcerative colitis (UC) OR inflammatory bowel disease (IBD).’ The search was restricted to English language original research including both full-text and abstract publications presenting TDM data from 1 January 2010 to 22 August 2022. After exclusion of duplicates, 146 articles were identified and imported into a systematic review platform (www.rayan.ai, accessed on 22 August 2022). Titles and abstracts were screened and approved independently by two reviewers (RDL and AJJ) to ensure relevance and availability of drug level data. Reference lists of selected articles were reviewed with additional publications selected as appropriate. Seven original publications were chosen for discussion (Figure 1), with a further three post hoc analyses included.

## 3. Pharmacokinetics of Subcutaneous CT-P13

The pharmacokinetics of intravenous infliximab are well described [49,50,51,52]. In short, administration via the intravenous route leads to early and rapid peak concentration followed by a steady decline to trough. Subcutaneously administered biologics have slower absorption, lower bioavailability, lower peak concentration and smaller differences between peak and trough concentrations. To date, there has only been one original, peer-reviewed published article defining the pharmacokinetics of subcutaneous CT-P13. Using patient data including a total of 2772 infliximab drug levels from the pivotal IBD CT-P13 Part 1 [40] and Part 2 [43] studies, Hanzel et al. constructed a population pharmacokinetic model incorporating the effect of body weight, anti-drug antibodies and serum albumin, given their known influence on clearance of intravenous infliximab [53]. The bioavailability of subcutaneous CT-P13 was reported as 79%, half-life 10.8 days and drug clearance estimated at 0.355 L/d in a typical patient weighing 70 kg, with a serum albumin of 44 g/L and no anti-drug antibodies. A prior subcutaneous CT-P13 Assessment Report by the EMA describing three separate population pharmacokinetic models based on clinical trials across healthy volunteers, rheumatological and gastroenterological indications calculated a bioavailability of between 58% and 72% [54]. It is important to note that many of these data are published in abstract form only or included in non-peer reviewed product reports. The estimated half-life and clearance of subcutaneous CT-P13 is comparable to findings from previous studies of intravenous infliximab in IBD, albeit with non-matched disease activity, weight, albumin and immunomodulator use between the models [49,50]. The calculated bioavailability of subcutaneous CT-P13 is comparable to that of adalimumab (64%) [55] and golimumab (52%) [56].

## 4. Impact of Dosing on Exposure-Response Relationship

Data from Part 1 of the pivotal subcutaneous CT-P13 RCT in patients with CD investigated the exposure-response relationship of 120 mg, 180 mg and 240 mg fortnightly subcutaneous CT-P13 doses in comparison to maintenance 5 mg/kg 8-weekly intravenous infliximab. At week 22–30, median subcutaneous infliximab drug levels incremented proportionally according to increasing subcutaneous dosing regimens (120 mg 2-weekly 13.3 µg/mL; 180 mg 2-weekly 19.9 µg/mL; 240 mg subcutaneous 2-weekly 26.5 µg/mL) and were significantly higher than those observed with intravenous infliximab (5 mg/kg 8-weekly 2.3 µg/mL) [40]. Similarly, in the Supplementary Materials of Part 2 of the pivotal RCT, mean trough levels at week 22 were higher in the 15 patients receiving 240 mg subcutaneous CT-P13 compared with 44 patients receiving 120 mg subcutaneous CT-P13 (mean [standard deviation; SD] 26.2 µg/mL [13.65] vs. 19.8 µg/mL [7.75], respectively) despite belonging to a higher weight category (80–115 kg vs. <80 kg, respectively) [43]. Despite allowing escalation to 240 mg 2-weekly from week 30 in patients with loss of response, data on the frequency of this event and subsequent changes in drug level are not presented. In contrast, escalating to a dose of 240 mg 2-weekly was strikingly effective in recapturing response in REMSWITCH, an observational, post-switch cohort of 133 patients [57]. Of the 22 patients who relapsed during the 6-month study period, 15 were escalated to 240 mg 2-weekly infliximab with recapture of clinical and combined clinical and biochemical remission in 93% and 80%, respectively. TDM data for this group of patients were not shown either at the time of relapse or following dose-intensification, as previously discussed [58].

In contrast, shortening the dose interval to weekly 120 mg subcutaneous CT-P13 may have less impact on drug levels. In a subgroup of 50 patients on prior dose-intensified intravenous infliximab from uncontrolled, real-world data by Smith et al., patients who switched to subcutaneous CT-P13 with a shortened dosing interval of 120 mg weekly had equivalent serum drug levels as patients switched to 120 mg 2-weekly at 3, 6 and 12 months despite not having worse baseline C-reactive protein (CRP) or faecal calprotectin (FCP) activity (median 16 vs. 16 µg/mL, *p* > 0.05 at all time points) [59]. Furthermore, receiving weekly vs. fortnightly dosing was not associated with trough infliximab drug levels on a linear regression analysis when controlled for multiple independent variables including disease activity, concomitant immunomodulator, anti-drug antibodies and body mass index (BMI) [59]. In summary, amongst patients requiring dose-intensified subcutaneous CT-P13, higher doses given fortnightly may achieve greater drug level increments than shortening the interval using standard 120 mg dosing, although more data are needed. The mechanism for this preliminary observation is unclear and not consistent with TDM data in adalimumab showing no difference in trough drug level between patients receiving 40 mg weekly and 80 mg fortnightly doses [60].

## 5. Comparing Drug Levels between Intravenous and Subcutaneous Infliximab

There are subcutaneous infliximab TDM data from a total of 465 individual patients across four published full-text articles [43,57,59,61] with additional data provided by 75 patients published in abstract or letter form [40,62,63] (Table 1). The majority of these studies compare drug levels between the two formulations taken at trough. However, trough drug levels between the two modes of administration are not directly comparable and do not reflect total drug exposure over a matched treatment period. This section will first outline available through TDM data, then contextualise these observations by discussing differing peak concentrations, relative drug level stability and comparable total drug exposure. Lastly, preliminary evidence supporting an exposure-response relationship will be presented.

Following intravenous induction at week 0 and week 2, receiving subcutaneous CT-P13 is associated with higher trough drug levels compared to continuing intravenous infliximab across the two available RCTs [40,43]. Data from Part 2 of the pivotal CT-P13 study in 131 patients show mean (SD) week 22 trough drug levels of 21.5 (9.9) µg/mL compared with 2.9 (2.6) µg/mL in the intravenous arm. When comparing geometric least squares mean (LSM), a more accurate estimate of true population mean, and adjusting for immunomodulator use, disease type, response status at week 6, and weight class, patients receiving subcutaneous CT-P13 had a trough drug level of 21.0 µg/mL compared to 1.8 µg/mL in the intravenous arm. As the lower bound of the 90% confidence interval for ratio of the geometric LSMs exceeded 80%, the primary outcome of pharmacokinetic non-inferiority of subcutaneous compared with intravenous CT-P13 was met. In addition, following switch to subcutaneous administration at week 30 in the original intravenous arm, the trough drug concentrations increased to comparable levels to those in the original subcutaneous arm. Similarly, when compared with pre-switch intravenous trough levels, both prospective and retrospective observational data confirm higher median trough drug levels following subcutaneous CT-P13 across time points ranging from 4 weeks to 6 months, except in patients requiring 10 mg/kg 4 weekly intravenous infliximab at the time of switching (Table 1).

Pharmacokinetic parameters that best predict optimal efficacy for infliximab are uncertain. As presented in Figure 2, possible predictors might be total drug exposure, maintenance of drug level stability, peak concentrations and trough concentrations, all of which differ between the two modes of administration. For intravenous infliximab therapy, TDM is performed at trough with a number of established concentration thresholds associated with varying depths of remission [46,47]. However, TDM performed earlier in the treatment cycle has also shown promising predictive potential, and the magnitude of these earlier drug levels may be important for severely active IBD [64,65]. Comparing only trough drug levels between subcutaneous and intravenous formulations does not accurately reflect differing peak concentrations. The most informative data regarding complementary pharmacokinetic parameters arise from Supplementary Materials of Part 2 of the pivotal RCT [43]. During the 8-week intensive TDM at steady state, the mean concentrations of subcutaneous CT-P13 are relatively stable when compared to the immediate peak and predictable decline of intravenous administration (Figure 3). Data generated by population pharmacokinetic modelling provide further detail. Whilst higher trough drug levels with subcutaneous infliximab are observed, the maximum peak drug level with subcutaneous infliximab is lower than with intravenous administration (mean 29.8 µg/mL vs. 105.6 µg/mL, respectively).

Comparable total drug exposure between subcutaneous and intravenous infliximab using the area under the curve (AUC) over an 8-week treatment period reflects the trade-off between trough and peak drug levels in the two formulations—graphically depicted in Figure 2. Using the same non-linear mixed-effect model, the mean AUC was slightly higher in the subcutaneous as compared with intravenous CT-P13 arms (35,467 μg·h/mL versus 28,284 μg·h/mL). Prospective, uncontrolled TDM data from 20 CD patients and a total of 120 drug levels taken across two fortnightly treatment cycles supports subcutaneous drug level stability both within and across cycles [61]. Similar to prior adalimumab data [66,67], a more stable steady state concentration-time profile offered by subcutaneous CT-P13 may allow greater flexibility with timing of TDM across the 14-day treatment cycle. However, more intensive pharmacokinetic analysis of adalimumab has demonstrated marked interpatient variability in subcutaneous absorption [68]. More data, ideally arising from a population pharmacokinetic analysis incorporating variables known to affect drug levels are required to translate these observations into clinical practice for subcutaneous CT-P13 [69]. Until these data emerge, we advise continuing to perform TDM of subcutaneous infliximab at trough where practicable.

Further studies are also required to determine target concentration thresholds for subcutaneous infliximab. In a post hoc analysis of 55 patients receiving subcutaneous CT-P13 in the Part 2 RCT, Ye et al. report preliminary data supporting an exposure-response relationship [70]. In this analysis, a higher proportion of patients with drug levels in the 4th quartile (≥26.7 µg/mL) achieved clinical remission and a faecal calprotectin ≤250 μg/g at week 54, as compared to patients with drug levels in the 1st quartile (<16.4 µg/mL) (79% vs. 46% and 91% vs. 62%, respectively). More data are needed to determine optimal drug level targets associated with depth of remission across a broader range of IBD phenotypes.

## 6. Predictors of Infliximab Drug Levels

Increasing body weight, presence of anti-drug antibodies, hypoalbuminaemia, absence of concomitant immunomodulation and increased disease activity are covariates that are associated with increase clearance of intravenous IFX, adalimumab and golimumab [49,50,51,52,56,68]. Similar data are accruing for subcutaneous CT-P13.

### 6.1. Body Weight

Evaluation of the effect of body weight on drug levels in the pivotal part 2 trial is limited by both exclusion of patients with obesity and the weight-based dosing regimen [43]. However, in their population pharmacokinetic model, Hanzel et al. reported body weight as a covariate affecting drug clearance by up to 43% between weights of 70 to 120 kg. In contrast, bioavailability of subcutaneous infliximab did not appear to be affected by body weight. Using Monte Carlo weight-based exposure simulations, receiving subcutaneous CT-P13 led to higher drug exposure in patients weighing 50 kg, comparable exposure in patients weighing 70 kg and lower exposure in patients weighing 120 kg in comparison to intravenous administration [53]. However, in the largest published real-world cohort of 181 patients post-switch from intravenous to subcutaneous IFX, trough drug levels were not affected by BMI, despite utilising non-weight-based dosing of subcutaneous CT-P13. Similarly, in a prospective drug sampling study evaluating TDM stability across the 14-day treatment cycle, Roblin et al. demonstrated that BMI had no association with low subcutaneous IFX drug levels (HR 0.83 95% CI 0.46–4.21; *p* = 0.69), although the number of patients who were overweight was small. Buisson et al. found that neither body weight nor BMI were associated with disease relapse amongst real-world switch data in 133 patients [57]. These findings may again be limited by low median BMI in the cohort. Clarification of the effect of body weight and body composition on pharmacokinetics and clinical outcomes is paramount given that obesity may modestly increase the odds of non-response to both fixed-dose and weight-based anti-TNF therapy [71], however, results are conflicting [72,73,74].

### 6.2. Serum Albumin

Hanzel et al. found that subcutaneous CT-P13 clearance was 30% greater when the serum albumin concentration was 32 g/L compared with that at 44 g/L [53]. Despite the limited pharmacokinetic understanding, hypoalbuminaemia is associated with lower intravenous infliximab drug levels [51]. Commonly proposed hypotheses include a correlation with increased inflammatory disease activity, protein catabolism and increased mucosal losses [75]. In healthy states, individuals with low albumin have lower neonatal Fc receptor (FcRn) activity and therefore accelerated clearance of IgG, including monoclonal antibodies [76,77]. How this relationship is altered in active IBD and the subsequent effect on monoclonal antibody clearance is unclear. In their comprehensive modelling study, Hanzel et al. found that no other biochemical parameters of disease activity (CRP, faecal calprotectin, platelet count) had a clinically relevant effect on drug clearance beyond the effect of hypoalbuminaemia [53]. In contrast, uncontrolled data from Roblin et al. showed no association between albumin and subcutaneous CT-P13 drug levels. However, numbers were small and all patients were in clinical and biochemical remission with a consequent homogenous and normal mean (SD) albumin of 39.6 (2.5) g/L at recruitment [61].

### 6.3. Immunomodulator Use

In IBD patients receiving intravenous infliximab, combination therapy with immunomodulators is associated with higher drug levels, less immunogenicity and subsequent greater disease control compared with those treated with anti-TNF monotherapy [66,78,79,80,81,82]. The benefit of concomitant immunomodulator use in patients receiving subcutaneous CT-P13 is less clear. In a post hoc analysis of 66 patients in the subcutaneous arm of the pivotal Part 2 CT-P13 trial, D’Haens et al. found comparable median (IQR) trough week 54 drug levels between those who received combination therapy with immunomodulators and those that received infliximab monotherapy (21.7 [19–25.3] vs. 20.8 [16.1–29.1] µg/mL, respectively) [83]. Similarly, there were no differences in clinical response rates (85% in combination therapy vs. 74% monotherapy; *p* = 0.3582) or development of neutralising anti-drug antibodies between the two groups (16% combination therapy vs. 7% monotherapy; *p* = 0.40) [83]. On a multivariate model evaluating a cohort of 181 patients switched to subcutaneous CT-P13 (58% on combination therapy), immunomodulator use was not associated with higher infliximab trough levels [59]. There was also no association between risk of disease relapse and immunomodulator use from the REMSWITCH cohort of 133 patients, of which 57% were receiving dose-intensified intravenous infliximab and 26% were on combination therapy at baseline. Further data, over a longer period of follow up are required to clarify the role of immunomodulators in subcutaneous CT-P13 therapy.

### 6.4. Immunogenicity

Immunogenicity to anti-TNFs is common, particularly in the first 12 months of therapy [66,80,81,84,85]. Detection of anti-drug antibodies is dependent on the type of laboratory assay, the dilution accuracy and the positivity thresholds. Drug-sensitive ELISA, electrochemiluminescence immunoassay (ECLIA) or radioimmunoassays can detect anti-drug antibodies only in the absence of drug, whereas drug-tolerant assays, such as homogenous mobility-shift assays (HMSAs) and newer ELISAs, can detect anti-drug antibodies in the presence of detectable drug [86,87,88]. Unsurprisingly, lowering the anti-drug antibody-positivity threshold increases the rate of detection of both transient anti-drug antibody and low-titre persistent, non-neutralising anti-drug antibodies [89]. The natural history and clinical relevance of these phenomena in comparison to the pharmacodynamic inactivation induced by neutralising anti-drug antibodies remain unclear [84,86,89,90]. Attention to laboratory methods must be made when interpreting immunogenicity data in IBD. In Part 2 of the pivotal subcutaneous CT-P13 RCT, total anti-drug antibodies and neutralising anti-drug antibodies were analysed using a drug-tolerant ECLIA platform with an affinity capture elution able to detect titres ≥25 and ≥1000 ng/mL, respectively. As Part 1 was only published in abstract form, there are no details on the assay used. In this study, 7/12 (58%) CD patients in the intravenous arm compared to just 3/30 (10%) in the subcutaneous arms developed anti-drug antibodies by week 30 [40]. In contrast, in the much larger Part 2 study, a similar proportion of patients in each arm converted to anti-drug antibody positive status over the 54 weeks (70% in subcutaneous vs. 64% in intravenous) but a smaller proportion of patients in the subcutaneous arm had positive neutralising anti-drug antibodies compared to the intravenous arm (18% vs. 37%, respectively; *p* = 0.019) [43]. No anti-drug antibody titres were presented in either study and there was no apparent impact of differing binary neutralising anti-drug antibody positivity rates on disease control and drug levels between the two groups. In four pharmacokinetic models examining intravenous infliximab in IBD patients, anti-drug antibodies have been observed to affect drug clearance by between 29% and 72% [49,50,52,91]. In their evaluation of drug levels from the pivotal Part 1 and Part 2 subcutaneous CT-P13 trials, Hanzel et al. estimated a congruent increase in clearance of 39% in patients with anti-drug antibodies. In the largest published real-world post-switch cohort of 181 patients, only 14 patients (8%) developed anti-drug antibodies [59]. Consistent with the prior modelling, anti-drug antibodies in these patients were strongly inversely associated with subcutaneous infliximab levels on multivariate analysis (OR −13.34, 95% CI −15.41x−11.33; *p* < 0.001) [59]. Interpretation of immunogenicity data from the available uncontrolled cohorts is limited by varying assay use (including across centres within the same study [59]) and the lack of comparator groups (Table 1).

Whilst the above results are preliminary, the potential for lower rates of immunogenicity reflects a promising theoretical advantage of subcutaneous infliximab. Traditionally, subcutaneously administered biologics were considered to be more immunogenic than intravenous therapy due to theoretical exposure to antigen-presenting cells within the epidermis and dermis [92], although objective evidence supporting higher antibody formation are conflicting [93,94,95]. There are several unproven hypotheses for subcutaneous infliximab being less immunogenic than intravenous administration. Low drug levels seen at the more pronounced concentration troughs with both maintenance [80,86] and episodic [96,97] intravenous therapy are associated with antibody formation. Comparing representative concentration-time curves, the drug level stability of subcutaneous dosing may avoid exposure to the more immunogenic concentration thresholds of intravenous therapy as depicted graphically in Figure 2. In addition, it has been suggested [43,59,98] that the higher circulating drug levels seen with subcutaneous CT-P13 may both reduce formation of immunogenic drug-antigen immune complexes and induce ’high-zone tolerance’. Whilst infliximab-TNF complexes have been demonstrated to drive anti-drug antibody formation [99], how this relates to the varying drug level exposure pattern of subcutaneous relative to intravenous infliximab is not clear. In ‘high-zone tolerance,’ exposure to high concentrations of an antigen may induce tolerance via blunting of the immune response [93,100,101,102]. Once again, it is not clear why this mechanism would be preferentially activated by the stable moderate drug levels of subcutaneous therapy and not the high peak concentrations of intravenous infliximab therapy. Further prospective, controlled trials with a longer duration of follow up are required to confirm a difference in anti-drug antibody formation, the antibody subtype and whether there are meaningful clinical consequences.

## 7. Conclusions and Future Directions

The available evidence suggests comparable efficacy and safety of subcutaneous infliximab for adult patients with UC or luminal CD, despite differences in pharmacokinetics such as bioavailability and the concentration-time profile. The economic advantages of biosimilar molecules are complemented by practical benefits such as patient convenience, reduced risk of in-hospital exposure to nosocomial infection, and alleviation of hospital resources and staffing pressures. However, potential disadvantages regarding adherence or inadequacy of drug storage require consideration. The most promising biological advantage of subcutaneous infliximab may be the stability of drug levels, as compared with the marked differences between peak and trough concentrations with intravenous therapy. Maintaining drug level stability may avoid the prolonged low trough levels associated with intravenous infliximab with a subsequent reduction in immunogenicity and a greater treatment persistence. On the other hand, the immediacy and magnitude of peak concentrations after intravenous infliximab may be the most relevant pharmacokinetic parameters to induce remission in highly active disease such as acute severe UC or severe, complex CD including perianal CD. Two ongoing large superiority RCTs in moderate-severe CD and moderate-severe UC have been powered to compare disease and safety outcomes (ClinicalTrials.gov Identifiers: NCT04205643 and NCT03945019, respectively) and may reveal the clinical significance of these differing pharmacokinetic profiles. Of importance, trough drug levels between the two modes of administration are not directly comparable and should not be considered in isolation. Future work should aim to clarify whether TDM has a role with subcutaneously administered infliximab and, if so, to define therapeutic concentration targets. Additional future directions include clarifying the role of immunomodulators, establishment of efficacy in paediatric IBD, examining adequacy of drug exposure for acute severe colitis and perianal disease and the optimal dosing regimen in patients previously requiring dose-intensified intravenous infliximab. More complete post-marketing data and real-world experience across the range of IBD phenotypes, distributions and severities will allow more precise positioning and optimisation of subcutaneous infliximab in the management of IBD.

## Figures and Tables

**Figure 1 jcm-11-06173-f001:**
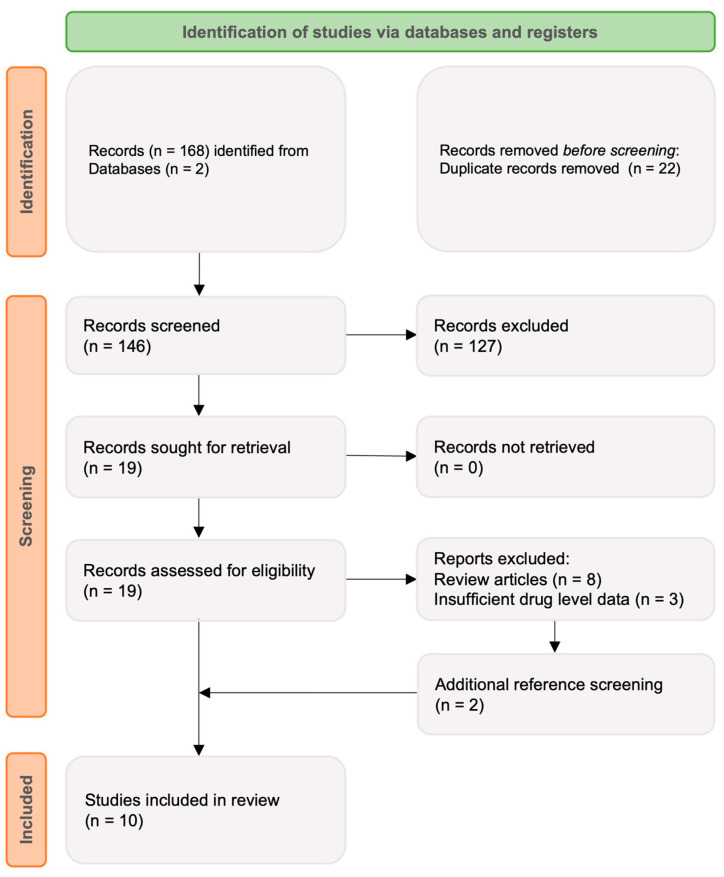
Search strategy outlining screening and eligibility assessment.

**Figure 2 jcm-11-06173-f002:**
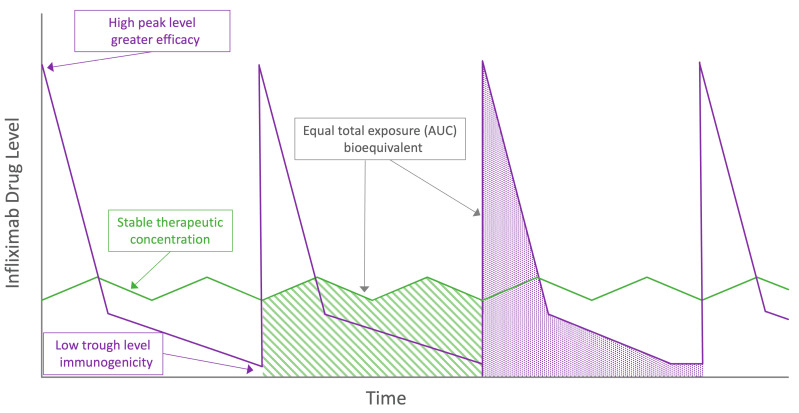
Visual representation of proposed theoretical pharmacokinetic advantages and disadvantages between intravenous (purple) and subcutaneous (green) infliximab. AUC = area under the curve.

**Figure 3 jcm-11-06173-f003:**
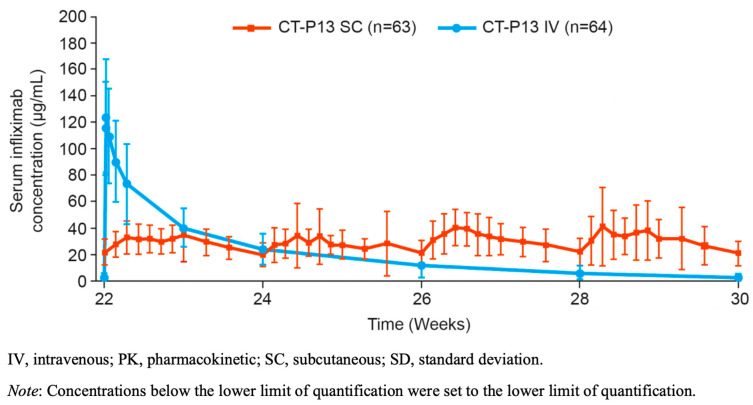
Mean (±SD) serum infliximab concentration for CT-P13 subcutaneous and CT-P13 intravenous arms during the more intensive 8-week sampling interval at steady state (PK monitoring period; PK population). Reprinted from: Gastroenterology, 2021; Schreiber S. et al. Randomized Controlled Trial: Subcutaneous vs. Intravenous Infliximab CT-P13 Maintenance in Inflammatory Bowel Disease (online Supplementary Materials, Figure S7, p. 31), Copyright (2021), with permission from Elsevier.

**Table 1 jcm-11-06173-t001:** Summary of original research reporting drug levels and disease outcomes in patients with inflammatory bowel disease (IBD) receiving subcutaneous infliximab (IFX; CT-P13) including full text and published abstracts.

Study	Design	Objectives	*n*	Characteristics	Drug Levels (µg/mL) and Anti-Drug Antibodies (µg/mL)		Disease Outcomes
					Intravenous (IV)	Subcutaneous (SC)	
Schreiber (2021) [43]	Multicentre (*n* = 50) randomised, open-label, non-inferiority trial.	Primary: to compare week 22 trough drug levels in patients exposed to IV or SC infliximab following IV induction. Secondary: to compare clinical outcomes between IV and SC infliximab.	131	41% CD, 60% UC 0% in remission Anti-TNF naïve Immunomodulator use: 44%	*Drug levels*: Mean (SD) W22 level: 2.9 (2.6) Adjusted geometric LSM W22 level: 1.8 *Anti-drug antibodies*: W22 ADA: 32 (49%) W22 nADA: 12 (19%)	*Drug levels*: Mean (SD) W22 level: 21.5 (9.9) Adjusted geometric LSM W22 level: 21 *Anti-drug antibodies*: W22 ADA: 21 (32%) W22 nADA: 4 (6%) *Laboratory assays*: Infliximab: ECLIA ADA: Drug-tolerant ECLIA with ACE	Comparable W30 and W54 clinical, biochemical, endoscopic response rates between IV and SC arms.
Smith (2022) [59]	Retrospective, multicentre (*n* = 3) cohort study.	Primary: to evaluate treatment persistence post-switch from IV to SC infliximab. Secondary: to compare clinical outcomes and drug levels between IV and SC infliximab.	181	64% CD, 33% UC, 3% IBD-U 87% in remission Prior IV infliximab: – 131 5 mg/kg q8W – 50 5 mg/kg q4 or q6W Immunomodulator use: 59%	*Drug levels*: Median (range) level: 8.9 (0.4–16)	*Drug levels*: Median level: 16 at 3, 6 and 12 months *Anti-drug antibodies*: Throughout study: 14 (8%) *Laboratory assays*: Drug-tolerant ELISA for infliximab levels plus free and bound ADA OR Drug-sensitive in-house ELISA for infliximab levels and ADA, dependent on centre.	Treatment persistence 92% No significant difference in clinical or biochemical activity between baseline and at 3, 6, or 12 months post-switch to SC infliximab.
Buisson (2022) [57]	Prospective, multicentre (*n* = 3) cohort study.	Primary: to assess clinical and pharmacological outcomes post-switch from IV to SC infliximab in IBD patients according to different IV infliximab regimens.	133	72% CD, 28% UC Perianal lesions (42%) 100% in remission Prior IV infliximab: – 44% 5 mg/kg q8W – 31% 10 mg/kg q8W – 14% 10 mg/kg q6W – 11% 10 mg/kg q4W Immunomodulator use: 26%	*Drug levels*: Median (IQR) baseline level: – 5 mg/kg q8W 4.7 (2.4–6.8) – 10 mg/kg q8W 7.2 (4.4–11.9) – 10 mg/kg q6W 8.1 (6.2–15.1) – 10 mg/kg q4W 18.5 (11.9–20) *Anti-drug antibodies*: 2 (2%) positive ADAs	*Drug levels*: Median (IQR) level at W16–24: – Prior 5 mg/kg q8W 15.1 (11.2–18.2) – Prior 10 mg/kg q8W 18.7 (8–20) – Prior 10 mg/kg q6W 14.3 (11.9–17.6) – Prior 10 mg/kg q4W 20 (17.7–20) *Anti-drug antibodies*: No positive ADAs *Laboratory assays*: Infliximab: ELISA ADA: drug-sensitive ELISA	By W16–24 a clinical or faecal calprotectin recurrence occurred in: – 10.2% 5 mg/kg q8W – 7.3% 10mg/kg q8W – 16.7% 10mg/kg q6W – 66.7% 10mg/kg q4W Intensification to 240 mg q2W, recaptured clinical remission in 93% (14/15).
Roblin (2022) [61]	Prospective, single centre cohort study.	Primary: to investigate the intra-individual variations of infliximab drug levels across and between 2 cycles of SC infliximab.	20	100% CD 100% in remission Immunomodulator use: 40%	*Drug levels*: Median (IQR) level: 3.9 (1.2–7.9) *Anti-drug antibodies*: No ADAs	*Drug levels*: Median (IQR) W8 level 11 (7.5–15.1) Similar level independent of sampling period (day 3–6, day 7–9, day 14). *Anti-drug antibodies*: No ADAs. *Laboratory assays*: Infliximab: ELISA ADA: drug-sensitive ELISA	No clinical relapse.
Abstracts and Letters:							
Schreiber (2018) [40]	Randomised, open-label controlled trial	Primary: find the optimal dose of SC infliximab in patients with active CD following IV induction at W0, W2 and randomisation 1:1:1:1 to: – IV 5 mg/kg q8W – 120 mg SC q2W – 180 mg SC q2W – 240 mg SC q2W Secondary: evaluate clinical outcomes and pharmacokinetics.	44	100% CD 0% in remission Immunomodulator use: not reported	*Drug levels*: Median W30 level (predicted interval 5th–95th percentile): 2.3 (0.1–8.6) *Anti-drug antibodies*: 7 (58%) positive ADAs	*Drug levels*: Median W30 TL (predicted interval 5th–95th percentile): – 120 mg SC: 13.3 (5.6–26.8) – 180 mg SC: 19.9 (8.4–40) – 240 mg SC: 26.5 (11.2–53.2) *Anti-drug antibodies*: 3 (10%) positive ADAs *Laboratory assays*: Not specified	Similar rates of clinical remission and response between SC and IV infliximab arms.
Chivato Martín-Falquina (2022) [62]	Retrospective, single-centre cohort study.	Primary: to report rates of remission and treatment persistence in IBD patients post-switch from IV to SC infliximab.	14	29% CD, 71% UC 100% in remission 79% prior intensified IV infliximab. Doses not specified Immunomodulator use: 64%	*Drug levels*: Median (IQR) level: 7 (2.4–10.5) *Anti-drug antibodies*: Not reported	*Drug levels*: W8 (IQR) level: 14.1 (IQR 12.2–22.7) *Anti-drug antibodies*: Not reported *Laboratory assays*: Not specified	Treatment persistence 93% 93% remained in clinical remission at 8 weeks.
Argüelles-Arias (2022) [63]	Retrospective, single centre, cohort study.	To assess efficacy and safety post-switch from IV to SC infliximab.	17	71% CD, 29% UC 100% in remission Immunomodulator use: 53%	*Drug levels*: Median (IQR) level: 6.1 (3.5–8.9) *Anti-drug antibodies*: Not reported	*Drug levels*: Median (IQR) W24 level: 19.9 (12.3–21.6) *Anti-drug antibodies*: Not reported *Laboratory assays*: Not specified	No clinical relapse but a reduced faecal calprotectin at W24 following switch to SC infliximab.

Abbreviations: UC (ulcerative colitis), CD (Crohn’s disease), IFX or CT-P13 (infliximab), SC (subcutaneous), IV (intravenous), ADA (antidrug antibody), nADA (neutralising ADA), SD (standard deviation), IQR (interquartile range), ELISA (enzyme-linked immunosorbent assay), ECLIA (electrochemiluminescence), ACE (affinity capture elution), W (week), q_W (_-weekly dosing).

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
