# Peer review of "Therapeutic Drug Monitoring of Subcutaneous Infliximab in Inflammatory Bowel Disease—Understanding Pharmacokinetics and Exposure Response Relationships in a New Era of Subcutaneous Biologics"

_jcm, 2022, doi:10.3390/jcm11206173_

Round 1
Reviewer 1 Report
I have read this comprehensive review article with much interest. This systematic review article is dealing subcutaneous infliximab in comparison with intravenous infliximab concerning TDM in IBD. Specially, following items are discussed; pharmacokinetics, impact of dosing on exposure-response relationship, comparing drug levels between intravenous and subcutaneous infliximab, predictor of infliximab drug levels, body weight, serum albumin, immunomodulator use, immunogenicity. All the sections are comprehensive and literatures indexed are quoted rightly.
Minor point; Figure 2 and Figure 3 are changed in right appearing order.
Author Response
Thank you for the review comments and drawing our attention to the Figure numbers. We have left the figure positions as they are but correctly adjusted their respective numbers.
Reviewer 2 Report
This is a well-written manuscript and provides important findings of subcutaneous infliximab for understanding management of patients with inflammatory bowel disease. I think this paper would be essential for future therapy. I have a few comments as follows.
In page 7 line 7, references should be inserted after the sentence of "Following intravenous induction at week 0 and week 2, ~ intravenous infliximab across the two available RCTs."
In page 9 line 10, I think that authors need to add when clinical remission and level of faecal calprotectin were evaluated in the Part 2 RCT reported by Ye et al.
In page 15 line 20, I think that development rate of anti-drug antibodies in each group of Part 2 CT-P13 trial reported by D'Haens et al. should be shown.
Author Response
Thank you for the comments and feedback.
In page 7 line 7, references should be inserted after the sentence of "Following intravenous induction at week 0 and week 2, ~ intravenous infliximab across the two available RCTs." - we have now inserted the references. Thanks
In page 9 line 10, I think that authors need to add when clinical remission and level of faecal calprotectin were evaluated in the Part 2 RCT reported by Ye et al. - thanks, this is useful information that we have now included.
In page 15 line 20, I think that development rate of anti-drug antibodies in each group of Part 2 CT-P13 trial reported by D'Haens et al. should be shown. - thanks, we have now included the rate of neutralising anti-drug antibodies.